# Acute Visual Impairment in a Patient with Parkinson’s Disease after Successful Bilateral Subthalamic Nucleus Deep Brain Stimulation with Low-Dose Levodopa: A Case Report

**DOI:** 10.3390/brainsci13010103

**Published:** 2023-01-05

**Authors:** Chao Zhang, Jinxing Sun, Zhenke Li, Na Liu, Chao Li

**Affiliations:** 1Department of Neurosurgery, Qilu Hospital of Shandong University, Jinan 250100, China; 2Institute of Brain and Brain-Inspired Science, Shandong University, Jinan 250100, China

**Keywords:** subthalamic nucleus, deep brain stimulation, Parkinson’s disease, visual symptoms

## Abstract

Background: Subthalamic nucleus deep brain stimulation (STN-DBS) is widely used for the treatment of primary motor symptoms in patients with Parkinson’s disease (PD). Further, recent evidence suggests that STN-DBS may relieve a few ophthalmic symptoms in PD, such as eye-blink rate and the flexibility of eye saccades. However, its exact effect on visual function remains unknown. Herein, we report the case of a patient with PD who underwent STN-DBS and experienced visual symptoms following levodopa reduction. Case presentation: A 63-year-old male patient with PD developed severe visual impairment after six months of high-frequency STN-DBS. His symptoms resolved after adjusting the levodopa dose prescribed to the patient. Conclusions: This case report suggests that DBS is beneficial in patients with PD in terms of eye-blink rate. However, the rapid reduction of medication after STN-DBS may lead to retinal atrophy and the shrinkage of vessel density in the ocular fundus. Thus, neurosurgeons should pay close attention to patients with visual symptoms when adjusting levodopa dosages.

## 1. Background

Parkinson’s disease (PD) is one of the most prevalent progressive neurodegenerative disorders characterized by a series of motor (tremor, rigidity, bradykinesia, and postural instability) and nonmotor symptoms (e.g., neuropsychiatric symptoms, cognitive decline, and visual disorders) [1]. Among the nonmotor symptoms, visual dysfunction is common in PD, which can have a significant impact on the activities of daily life and can increase the risk of falls and injuries [2]. The most commonly reported ophthalmologic symptoms include double vision, blurry vision, watery eyes, and visual hallucinations [3]. Deep brain stimulation (DBS) is widely used to treat advanced PD. However, the impact of DBS on visual disorders in PD has received little attention in both research and clinical practice. Only a few studies have reported that DBS affects eye movements [4]. Levodopa is commonly considered to protect retina morphology [5]. However, the levodopa reduction rate is usually close to 50% following DBS [6]. Herein, we report the case of a patient with PD who underwent DBS of the subthalamic nucleus (STN) and experienced visual symptoms following levodopa reduction.

## 2. Case Presentation

A 63-year-old man developed a gradual onset of rest tremor in his right arm and experienced a reduction in facial movements 12 years ago. The patient gradually noticed a static tremor in his right limb and a reduction in his facial movement range. Nine years ago, as his symptoms gradually progressed, he consulted a local neurologist and was diagnosed with PD. The patient was then started on Madopar (levodopa and benserazide hydrochloride tablets, 187.5 mg/day) and pramipexole (0.375 mg/day). The patient’s symptoms slowed, and he continued to visit his doctor, who made regular medication adjustments as needed. In the last half year preceding his visit to our hospital, the patient felt that the effects of the medication were shortening in duration, and he had increasing difficulties in walking and getting up. The patient was then referred to our hospital for DBS. When the patient was admitted, his medication regimen included Madopar (625 mg/day), pramipexole (0.75 mg/day), and entacapone (0.4 mg/day) (levodopa-equivalent dose (LEDD), 800 mg).

Neurological examination revealed facial hypomimia, bradykinesia, high muscle tone in the neck and extremities, and severe peak-dose dyskinesia. We also examined the visual symptoms of the patient using optical coherence tomography (OCT), fundus photography, and the Visual Impairment in Parkinson’s Disease Questionnaire (VIPD-Q) [7]. Results showed that the patient’s eye-blink rates were 13 times/min (OD) and 14 times/min (OS); the thicknesses of his retinal nerve fiber layer (RNFL) were 77 μm (OD) and 88 μm (OS) (normal range from 80 to 200 μm); his vessel percentage areas were 20.22 (OD) and 18.87 (OS); and his VIPD-Q score was nine (Table 1). After no contraindications for surgery were found, we performed STN-DBS, and postoperative CT/MRI imaging confirmed that the DBS leads were correctly placed in the STN (Figure 1A).

One month after surgery, the DBS parameters were selected as follows: right, C+1-, 60 μs, 130 Hz, and 1.8 V, and left, C+9-, 60 μs, 130 Hz, and 2.0 V, to which the patient’s motor symptoms appeared to respond well (UPDRS III: off stimulation, 47; on stimulation, 21). The patients’ eye-blink rates were 13 times/min (OD) and 14 times/min (OS), and his VIPD-Q score was seven at this time.

Six months after the commencement of the DBS treatment, the patient was bothered by the gradual worsening of his blurred vision and decreased strength when blinking. We conducted ophthalmologic examinations, and the results (Figure 1B) indicated that the thicknesses of the patient’s RNFL and vessel percentage area (VPA; normal range from 20 to 35) were dramatically decreased despite the improvement in eye-blink rate (Table 1, Figure 1C). The patient’s eye-blink rates were 15 times/min (OD) and 17 times/min (OS) (on stimulation) at this time. The RNFL thickness was 82 μm (OD) and 74 μm (OS), the vessel percentage areas were 19.25 (OD) and 14.37 (OS), and his VIPD-Q score was 16. One month after the operation, the patient’s medication was gradually reduced to Madopar (375 mg/day), pramipexole (0.375 mg/day), and entacapone (0.2 mg/day) (LEDD, 422.5 mg) within a three-week time period. DBS is seldom related to retinal morphology, and this patient did not benefit from having DBS turned off. We suspected that the occurrence of visual symptoms was related to the sharp reduction in levodopa, as previous studies have shown levodopa to have a protective effect on the retinal nerves [8,9]. Thus, we first restored the patient’s original levodopa dosage and reduced it gradually over time. The patient reported that his symptoms were alleviated with these treatment adjustments.

One year following surgery, the patient returned to our hospital for a further consultation, at which point his medication was restored to Madopar (600 mg/day) and pramipexole (0.75 mg/day) (LEDD, 675 mg). His ophthalmic symptoms had improved (Table 1, Figure 1C). His eye-blink rates were 18 times/min (OD) and 20 times/min (OS) (on stimulation) (Figure 2A); his RNFL thicknesses were 87 μm (OD) and 77 μm (OS) (Figure 2B); his vessel percentage areas were 24.82 (OD) and 22.51 (OS) (Figure 2C); and his VIPD-Q score was seven (Figure 2D). No obvious uncomfortable sensations had been experienced by the patient during the preceding year.

## 3. Discussion and Conclusions

Visual symptoms are common in PD, especially in advanced stages of the disease. These symptoms have been reported to be among the more notable nonmotor signs for the early diagnosis of PD, which often lead to functional impairment and a lower perceived quality of life [10]. Although visual dysfunction is a common nonmotor symptom in PD, the effect of STN-DBS treatment on this symptom remains unclear. According to some prior studies, DBS can relieve some ophthalmic symptoms such as eye-blink rate and the flexibilities of eye saccades [10]. This case indicated that DBS markedly improved eye-blink rates between on and off stimulation statuses.

The VIPD-Q is a newly developed questionnaire for evaluating visual impairment in PD. We introduced a score to screen for patients with visual dysfunction before DBS treatment. The VIPD-Q score of our patient showed a dramatic increase six months after surgery, which suggests that the patient’s visual symptoms may have been aggravated after adjusting the dosage of his levodopa medication. The improvement in RNFL thickness and vessel percentage area when levodopa was increased supports the results of previous research on levodopa’s ability to restore retinal morphology and visual function [11,12].

In summary, we presented the case report of a patient with severe visual impairment after STN-DBS. Although DBS had a consistent beneficial effect on the patient’s eye-blink rate, his RNFL and VPA decreased dramatically six months after surgery, and he reported significantly blurred vision. Visual symptoms were considered to correlate with the rapid reduction in his levodopa dosage, and the patient found that his symptoms were alleviated by restoring his original (pre-surgery) dose. To the best of our knowledge, this is the first report to correlate visual impairment with levodopa reduction after DBS. Thus, we conclude that postoperative drug reductions following DBS surgeries should be implemented slowly in PD patients with visual symptoms, and the levodopa dosage should be carefully reduced. Future large-scale studies are needed to confirm our findings and establish whether the favorable effects observed in our patient can be replicated to alleviate the detrimental effects of these clinical strategies or have an overall beneficial effect.

## Figures and Tables

**Figure 1 brainsci-13-00103-f001:**
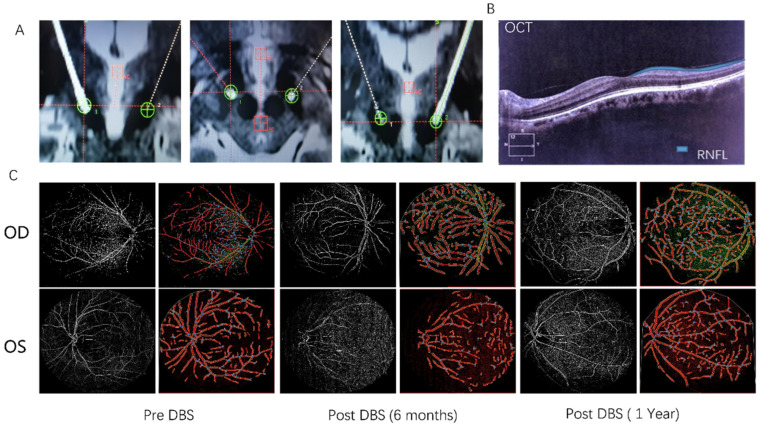
(**A**) The DBS contact locations. (**B**) The thickness of the patient’s retinal nerve fiber layer (RNFL) measured by OCT. (**C**) The vessel percentage areas of the patient before and after subthalamic nucleus deep brain stimulation (STN-DBS).

**Figure 2 brainsci-13-00103-f002:**
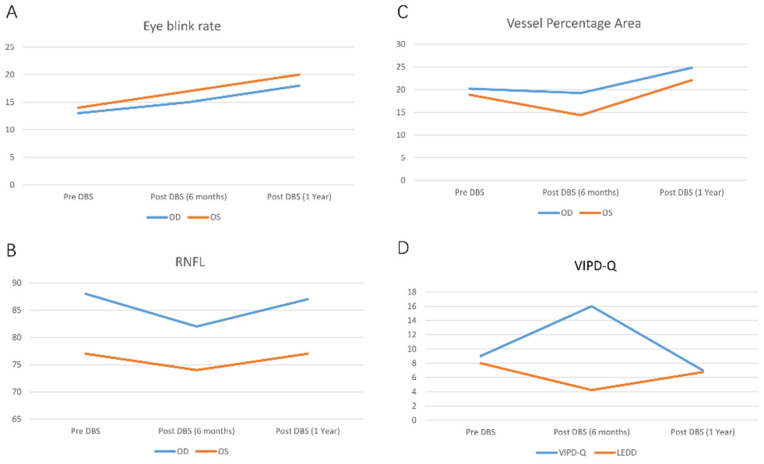
Fluctuation of ophthalmic characteristics pre- and post-STN-DBS. (**A**) Eye-blink rate. (**B**) RNFL. (**C**) Vessel percentage area. (**D**) VIPD-Q.

**Table 1 brainsci-13-00103-t001:** Patient’s visual symptom severity pre- and post-STN-DBS.

		Pre-DBS	Post-DBS (1 Month)	Post-DBS (6 Months)	Post-DBS (1 Year)
Eye-blink rate	OD	13	14 (on stimulation); 13 (off stimulation)	15 (on stimulation); 11 (off stimulation)	18 (on stimulation); 12 (off stimulation)
	OS	14	13 (on stimulation); 13 (off stimulation)	17 (on stimulation); 12 (off stimulation)	20 (on stimulation); 13 (off stimulation)
RNFL	OD	88	88	82	87
	OS	77	77	74	77
VPA	OD	20.22	20.12	19.25	24.82
	OS	18.87	18.86	14.37	22.1
VIPD-Q	-	9	9	16	7
LEDD	-	800	800	422.5	675

## Data Availability

All relevant raw data in the current study will be freely available to any scientist wishing to use them without breaching participant confidentiality for noncommercial purposes.

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
