# Peer review of "Acute Visual Impairment in a Patient with Parkinson’s Disease after Successful Bilateral Subthalamic Nucleus Deep Brain Stimulation with Low-Dose Levodopa: A Case Report"

_brainsci, 2023, doi:10.3390/brainsci13010103_

Round 1

Reviewer 1 Report

The authors present an interesting case of a patient who appears to have developed visual dysfunction upon a sudden reduction in dopaminergic medication dosage after DBS. This could be an impactful report, but it needs more context in the literature and more clarity in the writing.

Specific issues:

* The introduction needs much more information on visual dysfunction in PD. What types of dysfunction are common (similar to that seen in this patient?)? What is the reported impact of STN DBS and dopaminergic medication on visual function? Have others reported similar onset of visual issues following medication reduction? Importantly, what is the typical reduction rate of medication following DBS, and why do the authors think that this patient developed visual dysfunction when others do not?

* 1 month scores/measures (at the time of DBS programming) should be added to Table 1 if available

* on page 3, lines 73-74 ("At this time point, the medication treatment of the patient was reduced to...") are ambiguous - were the patient's medications changed at the 6 month time point (and then restored immediately), or had the medications been adjusted at the time of DBS programming and then were restored at the 6 month time point?

* on page 3, lines 76-77 ("...we doubt the occurrence of visual symptoms were related to the sharp reduction of medication...") is confusing (potentially a language issues) - it seems like the authors don't doubt this, they actually believe that the visual issues were caused by the medication reduction

* on page 3, lines 67-68 ("...the patient reported sudden blurred vision...") may be a language issue - were the symptoms sudden-onset? As in, did they appear suddenly, or did they gradually appear and were just reported at 6 months? If it was actually a sudden onset with weakness ("lacking in strength"), does that suggest an issue like stroke, hemmorrhage, etc?

* Please define acronym VPA

Author Response

Dear  Reviewer:

Thank you for your letter and for the reviewers’ comments concerning our manuscript entitled “Acute visual impairment in a patient with Parkinson’s disease after successful bilateral subthalamic nucleus deep brain stimulation: a case report and literature review” (ID: 2068778). Those comments are all valuable and very helpful for revising and improving our paper, as well as the important guiding significance to our researches. We have studied comments carefully and have made correction which we hope to meet with approval. Revised portion are marked in red in the paper. The main corrections in the paper and the responds to the reviewers’ comments are as flowing:

Responds to the reviewer’s comments:

The introduction needs much more information on visual dysfunction in PD. What types of dysfunction are common (similar to that seen in this patient?)? What is the reported impact of STN DBS and dopaminergic medication on visual function? Have others reported similar onset of visual issues following medication reduction? Importantly, what is the typical reduction rate of medication following DBS, and why do the authors think that this patient developed visual dysfunction when others do not?

Response: Thank you for the constructive suggestion. We are sorry for the lacking information in our paper. The most commonly reported ophthalmologic symptoms in PD included double vision, blurry vision, watery eyes, and visual hallucinations according to previous study. Some studies indicated that STN DBS could improve eye moments and Levodopa commonly is considered to protect the retina morphology. However, the typical reduction rate of levodopa is close to 50%. Ocular and visual disorders in Parkinson's disease were common but frequently overlooked symptoms in PD studies. To our knowledge, we are the first team reporting the visual impairment after levodopa reduction.

1 month scores/measures (at the time of DBS programming) should be added to Table 1 if available

Response: Thank you for your suggestion. As PD patients with STN-DBS usually maintain the levodopa dose and the stimulation usually did not show an immediate effect on visual function, we ignored to show the data in the old table 1. Thus, we added the original results in the new table 1.

on page 3, lines 73-74 ("At this time point, the medication treatment of the patient was reduced to...") are ambiguous - were the patient's medications changed at the 6 month time point (and then restored immediately), or had the medications been adjusted at the time of DBS programming and then were restored at the 6 month time point?

Response: We are sorry for not giving a realizing description of the levodopa. PD patients started reduction of levodopa 1 months after the implantation. The process usually takes 3-4 weeks to reduce the medication to a satisfying dose In this case, the patient took 3 weeks to reduce the medication to Madopar(375mg/day), pramipexole(0.375mg/day) and entacapone(0.2mg/day)(LEDD, 422.5) and maintained the combination until the time point(six months after the DBS surgery). After analyzing this case, we suggested the patients to restore the original dose to avoid further impairment of visual function.

* on page 3, lines 76-77 ("...we doubt the occurrence of visual symptoms were related to the sharp reduction of medication...") is confusing (potentially a language issues) - it seems like the authors don't doubt this, they actually believe that the visual issues were caused by the medication reduction

Response: We are sorry for the improper presentation in our paper. We had changed the sentence in this section to avoid the misunderstanding.

on page 3, lines 67-68 ("...the patient reported sudden blurred vision...") may be a language issue - were the symptoms sudden-onset? As in, did they appear suddenly, or did they gradually appear and were just reported at 6 months? If it was actually a sudden onset with weakness ("lacking in strength"), does that suggest an issue like stroke, hemmorrhage, etc?

Response: It was a language issue indeed. The visual dysfunction of this patient gradually appeared and was just reported at 6 months. We adjusted the words in this section to avoid further ambiguity.

Please define acronym VPA

Response: VPA is an abbreviation of “vessel percentage area”, which stands for the vessel density in retina layer. We added the explanation in abbreviation section and we are sorry for the mistake.

Considering the Reviewer’s suggestion, we have revised the content in the manuscript to meet the changes and it has been highlighted in red. We tried our best to improve the manuscript in accordance with the comments raised by the reviewer, as well as the manuscript was edited by one English native speaker.  Because the language editing does not change the content and framework of the paper, we thus did not mark these changes in red in the revised paper.

Special thanks to you for your good comments.

Reviewer 2 Report

In this manuscript, the authors report the occurrence of visual disturbances after STN DBS in a patient with Parkinson’s disease. The manuscript needs extensive linguistic revision by a native English speaker. On the scientific side, I think the title of the manuscript and some of the discussion are misleading.

What I understand from this case is that STN DBS improves the eye blinking rate, but does not improve other visual symptoms of PD. That is not surprising since many visual symptoms are related to retinal dopamine deficiency (Weil 2016: Visual dysfunction in Parkinson’s disease), which STN DBS does not restore. The fact that symptoms improved after increased dopamine medications support this point. In other words, there is no reason to think STN DBS itself causes visual impairment (somewhat implied in the title), but rather that lowering dopamine medication after DBS-related motor improvement can lead to increases in non-motor symptoms that are dopamine-dependent / DBS-independent.

Also, the title mentions a literature review. I would either remove that from the title or perform a more extensive, actual literature review. Many important publications on the topic have been omitted.

Author Response

Thank you for your letter and for the reviewers’ comments concerning our manuscript entitled “Acute visual impairment in a patient with Parkinson’s disease after successful bilateral subthalamic nucleus deep brain stimulation: a case report and literature review” (ID: 2068778). Those comments are all valuable and very helpful for revising and improving our paper, as well as the important guiding significance to our researches. We have studied comments carefully and have made correction which we hope to meet with approval. Revised portion are marked in red in the paper. The main corrections in the paper and the responds to the reviewers’ comments are as flowing:

*The fact that symptoms improved after increased dopamine medications support this point. In other words, there is no reason to think STN DBS itself causes visual impairment (somewhat implied in the title), but rather that lowering dopamine medication after DBS-related motor improvement can lead to increases in non-motor symptoms that are dopamine-dependent / DBS-independent.

Also, the title mentions a literature review. I would either remove that from the title or perform a more extensive, actual literature review. Many important publications on the topic have been omitted.

Response: Thank you for the tittle suggested. We are sorry for the improper description of our tittle and the precedent version of tittle has been replaced. STN-DBS only affected the blink rate of patient and the reduction of levodopa medication after the surgery was actually the real reason why patient suffered acute visual dysfunction.

Considering the Reviewer’s suggestion, we have revised the content in the manuscript to meet the changes and it has been highlighted in red. We tried our best to improve the manuscript in accordance with the comments raised by the reviewer, as well as the manuscript was edited by one English native speaker.  Because the language editing does not change the content and framework of the paper, we thus did not mark these changes in red in the revised paper.

Special thanks to you for your good comments.

Reviewer 3 Report

This is an interesting and important case study. Most of the reviewer comments are minor and grammatical in nature.

Line 20  when “adjusting”

Line27 replace usual with “common”

Line 32 replace occurred with “experiencing”

Line 36 capitalize “Two”

Line 37 and a “reduction” in facial

Line 40 symptoms “progressed”

Line 47 remove we evaluated and start sentence with “The neurological”

Line 55 replace we operated with “we performed the STN-DBS operation on him and post-operative…”

Line 62 remove “after” start sentence with “One month”

Line 68 replaced lacking with “decreased”

Line 75 “Since levodopa is reported to have”

Line 80 replace “1” with “One”

Line 93 “to be the most sensitive”

Line 96 replace in absence, with “is still unknown” or something similar

Line 95 define NMS (add this abbreviation to the abbreviation list)

Lines 103-105  “which suggests that the patient’s visual symptoms may have been aggravated after adjusting the amount of levodopa medication treatment. The increased use of levodopa improving the thickness of the RNFL and vessels percentage area support previous research about levodopa rescuing retinal morphology and visual function.

Line 108 replace constant improve effect with “beneficial effect”

Line 110 symptoms “were”

Line 113 rephrase “at the last time” it is unclear what this is referring to

Line 115 define what “it” means in this sentence, spell it out as it is unclear.

Author Response

Dear Reviewer:

Thank you for your letter and for the reviewers’ comments concerning our manuscript entitled “Acute visual impairment in a patient with Parkinson’s disease after successful bilateral subthalamic nucleus deep brain stimulation: a case report and literature review” (ID: 2068778). Those comments are all valuable and very helpful for revising and improving our paper, as well as the important guiding significance to our researches. We have studied comments carefully and have made correction which we hope to meet with approval. Revised portion are marked in red in the paper. The main corrections in the paper and the responds to the reviewers’ comments are as flowing:

Responds to the reviewer’s comments:

Response: We are sorry for the poor language of our manuscript. We had invited native speaker to refine this paper and also changed all the impropriate sentences according to your suggestion. We hope the flow and language level have been substantially improved. Thank you for your comments.

Considering the Reviewer’s suggestion, we have revised the content in the manuscript to meet the changes and it has been highlighted in red. We tried our best to improve the manuscript in accordance with the comments raised by the reviewer, as well as the manuscript was edited by one English native speaker.  Because the language editing does not change the content and framework of the paper, we thus did not mark these changes in red in the revised paper.

Special thanks to you for your good comments.

Round 2

Reviewer 1 Report

Thank you for the revision, I appreciate the author's clarifications. However, I think many of the clarifications are not represented in the manuscript, and would be helpful for readers.

Specifically,

* on page 3, lines 73-74 ("At this time, the patient’s medication was reduced to...") is still ambiguous to readers. Please clarify in the text that medication reduction was begun at 1mo post-op and reached the new levels at approximately 3 weeks later

* on page 3, lines 67-68 ("...the patient reported sudden blurred vision...") still sounds like the symptoms appeared suddenly. Please clarify in the text that the symptoms began gradually and were reported at 6 months.

Further, I think there still needs to be discussion about why this patient was so affected by medication reduction when so many others are not. Were medications tapered too quickly? Were there any other confounding issues for this patient?

Finally, several points of language in the text are still a bit confusing. Specifically:

*  on page 1, lines 41-42 ("A 63-year-old man presented with chronic diarrhea 12 years after a cephalosporin allergy. Two months after onset, the patient gradually noticed..."). The language and timeline here are confusing. The patient had an allergic reaction to cephalosporin, which caused chronic diarrhea for 12 YEARS? And the patient noticed PD symptoms 2 MONTHS after diarrhea onset or his initial visit to the hospital (again, 12 YEARS after the initial allergic reaction?). And then the patient first saw a neurologist 9 YEARS after first noticing PD symptoms?

Author Response

Thank you for your letter and for the reviewers’ comments concerning our manuscript. We have studied comments carefully and have made correction which we hope to meet with approval. Revised portion are marked in red in the paper. The main corrections in the paper and the responds to the reviewers’ comments are as flowing:

on page 3, lines 73-74 ("At this time, the patient’s medication was reduced to...") is still ambiguous to readers. Please clarify in the text that medication reduction was begun at 1mo post-op and reached the new levels at approximately 3 weeks later

* on page 3, lines 67-68 ("...the patient reported sudden blurred vision...") still sounds like the symptoms appeared suddenly. Please clarify in the text that the symptoms began gradually and were reported at 6 months.

Response: We are sorry for the poor language of our manuscript. We had changed all the impropriate sentences on page 3 according to your suggestion. We hope the clarifications have been substantially improved. Thank you for your comments.

* Further, I think there still needs to be discussion about why this patient was so affected by medication reduction when so many others are not. Were medications tapered too quickly? Were there any other confounding issues for this patient?

Response: We considered the primary cause of visual impairment in this patient is the medications tapered too quickly and add the expression in discussion part. Actually, we also observed some mild visual dysfunction in other patients with PD after medication reduction. We are now conducting a pilot study with large group to find the exact reason why some people suffered visual impairment and someone did not.

*  Finally, several points of language in the text are still a bit confusing. Specifically:

* on page 1, lines 41-42 ("A 63-year-old man presented with chronic diarrhea 12 years after a cephalosporin allergy. Two months after onset, the patient gradually noticed..."). The language and timeline here are confusing. The patient had an allergic reaction to cephalosporin, which caused chronic diarrhea for 12 YEARS? And the patient noticed PD symptoms 2 MONTHS after diarrhea onset or his initial visit to the hospital (again, 12 YEARS after the initial allergic reaction?). And then the patient first saw a neurologist 9 YEARS after first noticing PD symptoms?

Response: Thank you for your suggestion. We are sorry for the improper description. This patient suffered with diarrhea after a cephalosporin allergy 12 years ago, and two months after this diarrhea, the patient gradually noticed a static tremor in his right limb and a reduction in facial movement range. And then 3 years after that (9 years ago), he went to see a neurologist and diagnosed with PD.

Reviewer 2 Report

The authors have put effort to answer my comments and suggestions. The manuscript is greatly improved. However, the title is still misleading. I would suggest something along the lines of: “Abrupt reduction of dopamine medications following deep brain stimulation for Parkinson's disease may cause acute visual impairments: a case report”. 

Minor comments :

-       Line 22: STN-DBS may lead to retinal atrophy

I recommend the manuscript pending minor revisions.

Author Response

Thank you for your letter and for the reviewers’ comments concerning our manuscript. We have studied comments carefully and have made correction which we hope to meet with approval. Revised portion are marked in red in the paper. 

Response: Thank you for your suggestion. We added 'may' into the sentence in line 22.

Special thanks to you for your good comments.